# Besifloxacin Nanocrystal: Towards an Innovative Ophthalmic Preparation

**DOI:** 10.3390/pharmaceutics14102221

**Published:** 2022-10-18

**Authors:** José Izo Santana da Silva de Jesus, Felipe Rebello Lourenço, Kelly Ishida, Thayná Lopes Barreto, Valdir Carlos Avino, Edson dos Santos Neto, Nádia Araci Bou-Chacra

**Affiliations:** 1Department of Pharmacy, Faculty of Pharmaceutical Sciences, University of São Paulo, Professor Lineu Prestes Av 580, Cidade Universitária, São Paulo 05508-000, Brazil; 2Department of Microbiology, Institute of Biomedical Sciences, University of São Paulo, Professor Lineu Prestes Av 2415, Cidade Universitária, São Paulo 05508-000, Brazil; 3Department of Research and Development, Pearson Saúde Animal S.A., Rua Arizona 491, Edifício Itaporanga, 18° Andar, Conjunto 182, Cidade Monções, São Paulo 04567-001, Brazil; 4Ophthalmology Division, Clinical Hospital of the Faculty of Medicine, University of São Paulo, Avenida Dr. Eneas de Carvalho Aguiar, 255, Cerqueira César, São Paulo 05403-000, Brazil

**Keywords:** nanocrystals, conjunctivitis, besifloxacin, Povacoat^®^, fluoroquinolones

## Abstract

Bacterial conjunctivitis significantly impacts public health, including more than one-third of eye diseases reported worldwide. It is an infection caused by various aerobic and anaerobic bacteria and is highly contagious. Therefore, it has a high incidence of bacterial resistance to the antibiotics commonly used for treatment. Among the most recent antibiotics, besifloxacin is a fourth-generation fluoroquinolone antibiotic indicated exclusively for topical ophthalmic use. Due to its importance in treating bacterial conjunctivitis and its low solubility in water, limiting its efficacy, a nanotechnology-based drug delivery preparation was developed to overcome this hurdle. Besifloxacin nanocrystals were prepared by small-scale wet milling and response surface methodology, using Povacoat^®^ as a stabilizer. The particle’s average hydrodynamic diameter (Z-ave) was approximately 550 nm (17 times smaller than raw material), with a polydispersity index (PdI) of less than 0.2. The saturation solubility increased about two times compared to the raw material, making it possible to increase the dissolution rate of this drug substance, potentially improving its bioavailability and safety. The optimized preparation was stable under an accelerated stability study (90 days). The Z-ave, PZ, PdI, and content did not alter significantly during this period. Furthermore, the 0.6% m/m besifloxacin nanocrystals at the maximum dose and the Povacoat^®^ stabilizer did not show toxicity in *Galleria mellonella* larvae. The innovative ophthalmic preparation minimum inhibitory concentration (MIC) was 0.0960 µg/mL and 1.60 µg/mL against *Staphylococcus aureus* and *Pseudomonas aeruginosa*, respectively, confirming in vitro efficacy. Therefore, besifloxacin nanocrystals revealed the potential for reduced dosing of the drug substance, with a minor occurrence of adverse effects and greater patient adherence to treatment.

## 1. Introduction

Conjunctivitis refers to the inflammatory process of the conjunctiva and affects all age groups [1]. Bacterial conjunctivitis is responsible for more than one-third of eye complications worldwide, causing pain and discomfort to patients, decreasing their quality of life, and increasing health care costs [2,3,4,5]. Alfonso and colleagues [6] report that the costs of treating conjunctivitis ranged from 377 to 875 million dollars annually in the United States.

Bacterial conjunctivitis is highly contagious and can be caused by aerobic and anaerobic bacteria. It also includes other common species such as *Neisseria gonorrhoeae*, *Neisseria meningitides*, *Pseudomonas*, *Proteus*, and *Corynebacterium* [1,2,4]. Bacterial infections, if not treated properly, can have potentially serious consequences, including irreversible vision complications such as permanent loss [7]. Furthermore, in some cases, such as after glaucoma surgery, the contagion of bacterial conjunctivitis increases the patient’s risk of developing endophthalmitis [8]. This devastating complication can cause permanent vision loss [9,10].

In the case of more severe complications, such as the development of mucopurulent exudate, treatment with antibiotics should be immediate, including several classes of antibiotics, such as aminoglycosides, combinations of polymyxin B, macrolides, sulfonamides, and the fluoroquinolones [2,4]. The indication of the fluoroquinolone class as topical ophthalmic was only from 1990 onwards and brought significant therapeutic advantages, as it has a broad spectrum and low toxicity. However, many of these therapeutic classes are no longer effective or show bacterial resistance [11]. In this context, besifloxacin, a recent member of fourth-generation fluoroquinolones, has significant advantages. This antimicrobial was developed exclusively for topical ophthalmic use. Unlike other fluoroquinolones, besifloxacin is not used for treating other infections, and this drug substance is the only one in its class to present a lower risk of developing resistance [2,5].

The synthesis of besifloxacin was initiated by a Japanese company, and in 2009 it was approved by the Food and Drug Administration (FDA) for the treatment of bacterial conjunctivitis in the United States; two years later, it was approved by ANVISA in Brazil [12,13]. Besifloxacin shows low water solubility (0.143 mg/mL) and is marketed as besifloxacin hydrochloride by Bausch & Lomb Incorporated under the trade name Besivance^®^, 0.6% ophthalmic suspension [5,14,15].

Conventional ophthalmic preparations (drops) are rapidly removed from the ocular surface due to eye protection mechanisms. Therefore, the retention time of drugs in the eye is minimal with consequently low bioavailability, generally less than 5% [16,17]. A different nano-based besifloxacin delivery was developed to overcome these challenges [17,18,19,20]. However, there are no besifloxacin nanocrystals reported in the literature. Nanocrystal increases the surface area to volume ratio of the drug substance, enhancing its saturation solubility, dissolution velocity, and adhesiveness when compared to the drug on the micrometric scale, with a consequent increase in bioavailability [21,22,23,24,25]. Given these benefits, the global nanotechnology market for drug delivery is projected to reach US$ 164.1 billion by 2027 [26], showing that the global nanotechnology market for drug delivery is projected to reach US$ 164.1 billion by 2027. Nanocrystals alone should reach US$ 83.1 billion in this period, with a growth of 21.9% for the next years.

In general, drug nanocrystals are prepared in dispersion media with the addition of stabilizers (surfactants and polymers), resulting in a colloidal state referred to as nanosuspensions [21,22]. The choice of stabilizer is complex and critical in the development stage of nanocrystals. Design space, a strategic tool in the QbD concept, allows for understanding and establishing multivariate combinations and interactions of input variables for the proper development of the product or process [27,28].

Therefore, this study aims to develop besifloxacin nanocrystals using a spatial design approach with subsequent physicochemical characterization, in vivo toxicity (*Galleria mellonella model*), and in vitro efficacy by determining the minimum inhibitory concentration against two main bacteria causing conjunctivitis.

## 2. Materials and Methods

### 2.1. Materials

Besifloxacin (99.93%, MW 430.301 g/mol, Log P 0.54, water solubility: 0.143 mg/mL) was purchased from Jinan Shengqi Pharmaceutical Co., Ltd. (Jinan China). Povacoat^®^ (POVA) was donated from DAIDO Takeshima Nishiyodogawa-Ku, Osaka, Japan). Polysorbate 80, cetylpyridinium chloride (CPC), benzalkonium chloride (BAC), benzyldimethylhexadecylammonium chloride 16-BAC), and chitosan (CS) were purchased from Sigma-Aldrich (St. Louis, MO, USA). Poloxamer 188 (P188), Poloxamer 407 (P407), Poloxamer HS 15 (HS 15), tocopherol polyethylene glycol succinate (TPGS), and Kollicoat^®^ IR (KLT) were purchased from BASF (Ludwigshafen am Rhein, Germany). Triethylamine, phosphoric acid, acetonitrile, and phosphate buffer solution pH 7.0 (Certipur^®^) were purchased from Merck (Darmstadt, Germany). The product Besivance^®^ (0.6% ophthalmic suspension) (Bausch & Lomb Incorporated, Bridgewater, NJ, USA) was obtained from a local market. Milli-Q^®^ Integral 10 ultrapure water (Merck KGaA, Darmstadt, Germany) was used for analytical purposes.

### 2.2. Drug Analysis

#### 2.2.1. Particle Size Determination by Laser Diffraction

The particle size of besifloxacin (raw material) was determined and compared with the commercial product (Besivance^®^), for which the Beckman Coulter™ LS 13 320 laser diffraction particle size analyzer equipped with a ULM module (Universal Liquid Module) was used (Beckman Coulter Inc., Brea, CA, USA). A suspension of the drug was prepared in phosphate buffer pH 7.0, and the commercial product (Besivance^®^) was used directly for analysis. Results represent the average of three readings.

#### 2.2.2. Determination of Minimum Inhibitory Concentration (MIC)

The MIC of besifloxacin (raw material) against *Staphylococcus aureus* ATCC^®^ 23235™ and *Pseudomonas aeruginosa* ATCC^®^ 9027™ were determined by the microdilution method. Posteriorly, tubes containing the serial dilutions (0.125 to 60 µg/mL) were incubated in an oven at 37 °C for 24 h. The determination of the inhibition and/or growth of microorganisms was performed by spectrophotometer [29].

### 2.3. Preparation of Besifloxacin Nanocrystals Using Small-Scale Wet Bead Milling

#### 2.3.1. Exploratory Tests for Stabilizer and Statistical Variables Selection

The besifloxacin nanocrystals were obtained using a small-scale top-down approach by wet bead milling, according to Romero, Keck, and Müller [30]. The system was composed of three magnetic stirring bars (5 mm each) arranged on top of each other in a 10 mL glass vial, stirred by a magnetic stirring plate (IKA^®^-WERKE, Staufen im Breisgau, Germany). In the first exploratory assay, the proposed formulation consisted of 0.6 wt% of besifloxacin (BSF) and 0.6 wt% of conventional stabilizers (1:1 wt ratio). In the second part, the same drug concentration was maintained (0.6 wt%); however, new stabilizers and different proportions were tested. All formulations had their total weight adjusted to 10 g with buffer solution pH 7.0 (dipotassium hydrogen phosphate/disodium hydrogen phosphate). Additionally, 30% of the vial volume was filled with 3.0 g (*w*/*w*) zirconium beads (size of 0.1 mm) (Retsch, Germany).

Initially, each tested stabilizer was weighed and transferred to the vial containing half of the planned buffer solution, and the system was stirred for 10 min to ensure total solubilization. In sequence, the besifloxacin was weighed and added to the formulation and the remaining buffer solution to complete 10 g of nanosuspension. The milling process was at 1200 rpm for 4 days for all stabilizers. The measurements of average hydrodynamic diameter (Z-ave) and polydispersity index (PdI) were determined within 24 h of the milling process.

The formulations whose stabilizers allowed particle size reduction to nanometer-scale were kept at room temperature. Z-ave and PdI measurements were determined after 7 and 15 days to verify the maintenance of particle size stability. Additionally, as a part of the third preliminary study, the processing time (24 and 96 h) and the number of magnetic bars (2 and 3) were tested with the stabilizer Povacoat^®^, chosen for the optimization step.

#### 2.3.2. Nanosuspension Optimization Process Applying Box–Behnken Experiment Design

The Box–Behnken design refers to a response surface design without considering the built-in full factorial or fractional design (Minitab^®^ 18). For this model, at least three factors were considered. In this sense, the exploratory studies allowed to establish the agitation speed as a critical parameter of the process, as well as the stabilizer (Povacoat^®^), the drug (besifloxacin), and their concentrations, to optimize the formulation. These independent variables were selected for the experiment, as follows in Table 1. Thus, this experiment was used to evaluate the responses of the dependent variable, average hydrodynamic diameter (Z-ave).

#### 2.3.3. Mathematical Model Verification

The mathematical equation derived from the response surface analysis determined the besifloxacin nanosuspensions formulations for the statistical model verification. These formulations were prepared using the conditions described in this section on the nanosuspension optimization process applying a Box–Behnken experiment design. The Z-ave measurement was determined at the end of this period.

### 2.4. Dynamic Light Scattering and Zeta Potential

The nanocrystals Z-ave and polydispersity indexes (PdI) were determined by photon correlation spectroscopy using Zetasizer Nano SZ90 (Malvern Instruments, Malvern, UK). Light scattering was monitored at 25 °C at an angle of 90°. The samples were previously diluted in the besifloxacin saturated solution (ratio of 1:24) for a total of 1 mL, considering an instrument sensitivity range. The measurements used disposable polystyrene cells. The analysis was performed once for each nanosuspension, and the software performed three runs.

The ZP was carried out by electrophoretic light scattering using the same equipment. Three measurements were taken for each sample, each comprising 14 accumulations. The measurements were taken at 25 °C before dilution with buffer solution pH 7.0 (conductivity = 2238 µS/cm). The applied field force was 20 V/cm, and the results were obtained from a Henry equation [31].

### 2.5. Distribution and Particle Size by Laser Diffraction (LD)

The distribution and particle size of the besifloxacin nanosuspension (0.6 w%) were performed by low-angle laser scattering using the Mastersizer 2000 equipment (Malvern Instruments, Malvern, UK), equipped with a Hydro 2000MU module (Malvern Instruments), Malvern, UK). The equipment was programmed for the wet method with a measurement range from 50 nm to 1 mm, using pH 7.0 phosphate buffer for dispersion. Granule refractive index: 1.67. The results were presented based on the volume distribution (D10, D50 and D90), which indicates the percentage of existing particles below the given size (nm).

### 2.6. Density, Viscosity, and pH

The density was determined using a DMA 4500 M densimeter (Anton Paar GmbH, Graz, Austria). For the measurement, the sample was placed directly in the equipment, in enough quantity to fill the hose/channel. The density was determined concerning water at 25 °C.

The viscosity was performed using an LVDV-I PRIME viscometer (Brookfield, Middleboro, MA, USA), equipped with an LV-3 spindle (63), and the speed was 50 rpm. For testing, 15 mL of the nanosuspension was transferred to the SC4-13RPY sample container (Brookfield, Middleboro, MA, USA).

The pH was determined with an Orion™ Versa Star Pro™ potentiometer (Thermo Fisher Scientific, Waltham, MA, USA). All these analyses were performed directly on besifloxacin nanosuspension samples, without dilution, in triplicate at 25 °C.

### 2.7. Osmolality

The osmolality of the optimized formulation was determined using the A₂O Advanced Automated Osmometer (Advanced Instruments, Norwood, MA, USA). For this, 1 mL of the nanosuspension was transferred to a glass tube and placed on the disk of the equipment that performs the injection automatically. In this equipment, measurements were determined from the freezing point depression of the sample. The sample injection volume was 100 µL, and the temperature range for determining the freezing point was –40 °C to +45 °C. Measurements were performed in triplicate.

### 2.8. Validation of the Method by High-Performance Liquid Chromatography (HPLC)

The reference method was developed by COSTA and colleagues using the commercial product Besivance^®^ (0.6% ophthalmic suspension) [12]. The following performance characteristics were determined: linearity, selectivity, precision, accuracy, and system suitability [32,33]. The excipients of the commercial product were measured during a validation study (selectivity) in the reference method [12]. Additionally, a placebo was prepared to contain Povacoat^®^ (9 mg/mL), glycerol (10 mg/mL), and benzalkonium chloride (1 mg/mL) to evaluate possible interferences in the analytical responses of this method.

#### 2.8.1. HPLC System and Apparatus

The Dionex™ UltiMate™ 3000 UHPLC^+^ liquid chromatography (Thermo Fisher Scientific, Waltham, MA, USA) was used, consisting of a DGP-3600 quaternary pump, ACC-3000 autosampler (with sample thermostat), ACC-3000 column compartment, and DAD detector 3000 (RS) equipped with Chromeleon™ software version 6.80. The C18 Eclipse Plus 150 mm × 4.6 mm × 3.5 µm column (Agilent Technologies, Santa Clara, CA, USA) was used. The mobile phase consisted of a mixture of 0.5% (*v*/*v*) triethylamine solution (pH adjusted to 3.0 with dilute phosphoric acid (10% *v*/*v*)) and acetonitrile (74:26 *v*/*v*) at a flow rate of 1.0 mL/min. The injection volume of the solutions was 20 µL, with a column temperature of 25 °C, UV detection at 295 nm, using a diode matrix (DAD) [12].

#### 2.8.2. Solutions

Stock solutions of the standard (SQR) and samples containing 500 µg/mL of besifloxacin were prepared. From these solutions, aliquots were diluted using the mobile phase to obtain adequate concentrations to determine the performance characteristics of the method. The solutions were transferred to a 2 mL vial using a Millex^®^-HV-PVDF filter 13 mm in diameter, 0.45 µm pore size, and then analyzed. Additionally, a filter test was performed to assess the compatibility and possible interference in the analytical responses [34].

The density of the besifloxacin nanosuspension (0.6 wt%) was determined as 1.0081 g/mL. This result was considered for weighing the aliquot of the nanosuspension. Afterwards, a sample solution was prepared with a concentration of 50 µg/mL of besifloxacin.

### 2.9. Scanning Electron Microscopy (SEM)

The raw material (besifloxacin hydrochloride) and nanocrystals (lyophilized nanosuspension) were analyzed using a Quanta™ 650 FEG scanning electron microscope (SEM) (FEI Company, Hillsboro, OR, USA). The samples were plated with platinum using a metallizer MED-020 high-vacuum coating system (Baltic, Pfäffikon, Switzerland). This metallization (the layer between 10–15 nm thick) guarantees the conduction of the electron beam in the analyzed sample. For analysis, 10 kV of electron acceleration voltage was applied. The raw material besifloxacin was analyzed at 5000 to 30,000 times magnification, and the nanocrystal was analyzed at 5000 to 100,000 times magnification.

### 2.10. Lyophilization

The lyophilized samples were used for the analyses that needed dry samples (lyophilized powder), which was obtained by drying the nanosuspension using the lyophilization process using a Christ Alpha 2-4 LSC (Martin Christ Gefriertrocknungsanlagen GmbH, Osterode am Harz, Germany). For this, 1.5 mL aliquots of nanosuspension were frozen at −45 °C with a pressure of 0.120 mbar for 48 h. After lyophilization, the besifloxacin nanocrystals were resuspended in the same volume of pH 7.0 buffer (proportional to the volume of the initial formulation), and the Z-ave, IP, and PZ were determined to verify the efficiency of lyophilization.

### 2.11. Thermal Analysis

Differential scanning calorimetry (DSC) analysis was performed using DSC 7020 equipment (Hitachi High-Tech Corporation, Tokyo, Japan). All measurements were performed under a dynamic nitrogen atmosphere (50 mL min^−1^). The weight of the samples was approximately 2 mg. DSC curves were obtained by heating at a rate of 10 °C min^−1^ from 25 to 300 °C. As a reference, an empty sealed aluminum crucible was used.

The thermogravimetry and derived thermogravimetry (TG/DTG) curves were obtained using an STA 7200 analyzer (Hitachi High-Tech Corporation, Tokyo, Japan), with a sample weight of approximately 2 mg in the temperature range from 30 to 600 °C, at a rate of 10 °C min^−1^ under a dynamic nitrogen atmosphere (100 mL min^−1^). The lyophilized nanosuspension, besifloxacin (raw material), Povacoat^®^, benzalkonium chloride, and a mixture of both (same proportion of the formulation) were characterized.

### 2.12. X-ray Diffraction (XRD)

X-ray diffraction analyses were performed using a Rigaku Miniflex 600-C diffractometer (Rigaku Corporation, Tokyo, Japan). For this, the samples were ground to obtain a fine powder. Next, the powder was spread evenly in the rotating glass sample holder (zero background) with the aid of a glass slide. The equipment was configured in reflection mode with a scan range from 2.3° to 50° 2Θ, and a speed of 10°/min (0.02° step) was applied. The diffractograms were generated by the SmartLab Studio II software (Rigaku Corporation, Tokyo, Japan).

### 2.13. Saturation Solubility

The saturation solubility was evaluated using the shake-flask method [35], using a Tecnal TE 4080 (Tecnal, São Paulo Brazil) stirring oven at 37 ± 1 °C with stirring 150 RPM. For the study, the following media were considered: ultrapure water Milli-Q^®^ Integral 10 (Merck KGaA, Darmstadt, Germany), pH 4.0 phosphate buffer (citric acid/sodium hydroxide/hydrogen chloride), Certipur^®^ (Merck KGaA, Darmstadt, Germany), phosphate buffer pH 6.8, and phosphate buffer pH 7.0 (disodium dipotassium hydrogen phosphate hydrogen phosphate) Certipur^®^ (Merck KGaA, Darmstadt, Germany).

To prepare the saturated solution of the drug, 250 mg of besifloxacin (raw material) was weighed and transferred to an Erlenmeyer flask containing 250 mL of the previously described media (concentration 1 mg/mL). Similarly, 4130 mg of the besifloxacin nanosuspension (equivalent to 25 mg of besifloxacin, density 1.0081 g/mL) was transferred to Erlenmeyer of 25 mL containing 25 mL of the media (concentration 1 mg/mL). For these preparations, the declared solubility of besifloxacin of 0.143 mg/mL was considered [5].

The samples were incubated, and after 24 and 48 h, an aliquot of 1 mL was removed, filtered Millex^®^-HV-PVDF 13 mm in diameter, with a pore size of 0.45 µm. From this filtrate, solutions with a theoretical concentration of 50 µg/mL were prepared using the respective media (ultrapure water, pH 4.0 buffer, pH 6.8 buffer, and pH 7.0 buffer). Afterwards, the sample solutions were transferred to a 2 mL vial using Millex^®^-HV-PVDF filters 13 mm in diameter, pore size 0.45 µm, and quantified in triplicate (three injections) by HPLC.

### 2.14. In Vivo Toxicity

Larvae of *Galleria mellonella* ranging from 2.0 to 2.5 cm in length and 150 to 200 mg in weight were used to determine the toxicity of besifloxacin nanocrystals and the Povacoat^®^. Doses of 500 mg/kg of besifloxacin nanocrystals (0.06 mg/larvae) and 750 mg/kg of Povacoat^®^ (0,09 mg/larvae) were injected (10 µL) in the last left pro-leg of larvae (*n* = 16 larvae per group). In addition, sterile phosphate-buffered saline (pH 7.0) was also injected into a larval group to control mechanical injury and vehicle. The larvae were maintained at 37 °C, and the survival of all groups was monitored daily for 5 days [36].

### 2.15. Determination of Minimum Inhibitory Concentration (MIC)

The minimum inhibitory concentration (MIC) was determined for besifloxacin hydrochloride (raw material) and for besifloxacin nanosuspension 0.6 wt% against *Staphylococcus aureus* and *Pseudomonas aeruginosa*. Similarly, a serial dilution method was used, according to item 2.2.2 [29].

### 2.16. Stability Study

The accelerated stability study of the besifloxacin nanosuspension was conducted in a Climacell Eco-Line climate chamber (MMM Medcenter Einrichttungen GmbH, Planegg, Germany) considering the main regulatory guidelines [37,38]. For this purpose, three batches of the formulation were prepared, and the samples were transferred to type I amber glass vials, closed with butyl caps, sealed with an aluminum seal, and conditioned for accelerated stability for 90 days at the condition: 40 °C ± 2 °C/75%RH ± 5%RH.

Monthly, Z-ave, PdI, ZP, pH, and aspect were determined. In addition, the besifloxacin content was determined at time 0 and after 90 days. Analysis of variance (ANOVA) was performed using Excel software.

## 3. Results and Discussion

### 3.1. Drug Analysis

#### 3.1.1. Particle Size Determination by Laser Diffraction

The initial particle size distribution for besifloxacin hydrochloride (raw material) and commercial product (Besivance^®^ 0.6% ophthalmic suspension) was determined as a preliminary analysis. The particle size distribution (Figure 1) shows the same bimodal formation for both product and raw material samples. The central peak presents the most extensive distribution of particles, with a diameter 9.33 ± 3.22 µm and 10.62 ± 1.76 µm for raw material and Besivance^®^, respectively. In addition, the D90 shows that 90% of the particles have a diameter smaller than 13 µm for both samples.

Therefore, the raw material besifloxacin hydrochloride is presented on a micrometric scale, close to the reference values (commercial product). Drug nanocrystal prepared using small-scale wet bead milling can overcome the low water solubility hurdle of the raw material, which potentially improves drug bioavailability justifying the present study.

#### 3.1.2. Determination of Minimum Inhibitory Concentration (MIC)

Likewise, the in vitro antimicrobial activity of besifloxacin hydrochloride (raw material) and Besivance^®^ were determined using microorganisms that frequently cause bacterial conjunctivitis.

MIC for *Pseudomonas aeruginosa* against besifloxacin hydrochloride and the product Besivance^®^ was 1.58 and 0.99 µg/mL, respectively. The values reported in the literature are between 0.5 and 4 µg/mL [8,29,39]. Against *Staphylococcus aureus*, the MIC was 0.0957 and 0.0959 µg/mL for besifloxacin hydrochloride and the product Besivance^®^, respectively. The literature reported values between 0.008 and 8 µg/mL [8,29,39]. Thus, the raw material met the requirements for use in the drug nanocrystal development process.

### 3.2. Preparation of Besifloxacin Nanocrystals Using Small-Scale Wet Bead Milling

#### 3.2.1. Exploratory Tests for Stabilizer and Nanosuspension Optimization Process Applying Box–Behnken Experiment Design

The choice of stabilizer is complex and critical in the development stage of nanocrystals. In the present study, 11 stabilizers were tested to obtain nanocrystals. These stabilizers were compared considering their efficiency in particle size reduction and the preliminary results for their physical stability. For this, 24 formulations were evaluated with a single stabilizer or combinations, divided into three stages.

In the first exploratory study, the stabilizers conventionally used to obtain nanocrystals were tested (Appendix A), and no significant size reduction or no physical stability was observed. In the second part, formulations with new and promising stabilizers were prepared (Appendix A). Therefore, a study was carried out to evaluate the stability of the formulations with the stabilizers that reduced the size of the particles to the nanometric scale in the first and second exploratory studies (Appendix A).

From the exploratory study, it was found that the stabilizer Povacoat^®^ allowed us to obtain an innovative formulation for treating bacterial conjunctivitis. Figure 2 shows formulas F1, F3, F7, and F8 selected from preliminary studies. However, after 15 days, F1 containing Povacoat^®^ presented physical stability. Developed as a pharmaceutical excipient, Povacoat^®^, with consolidated use for coating and wet granulation processes, has a pKa of 3.56 in strong acid and 9.5 in strong base, with a load of 0.03 in pH 6.5. This copolymer is established as a stabilizing agent for obtaining nanocrystals, preventing aggregations, and improving the saturation solubility of nanosuspensions with drugs with nanometer-scale particles [28,40,41,42,43,44,45,46].

The concept of design space has been consolidated as a fundamental step in pharmaceutical development. This approach allows the understanding and establishing multivariate combinations and interactions of input variables for the proper development of the product or process [28,47,48]. The main variables found in exploratory studies were considered in this optimization.

The concentration of Povacoat^®^ (1.0 wt%) obtained in the preliminary investigation was considered to establish the maximum concentration of this stabilizer for the experimental design (+1). The central level (0) was 0.8 wt% and the minimum concentration (−1) established was 0.6 wt%, with the prospect of obtaining the same concentration concerning the drug (1:1). This concentration was also tested in preliminary trials. Although nonionic stabilizers are less toxic than cationic, these should be used with as little as possible concentration to avoid patient risks [48].

For drug substance concentration, the following levels were considered: (−1), central (0) and (+1), 0.4, 0.6, and 0.8 (wt%), respectively, aiming to evaluate a possible synergistic effect in the reduction of Z-ave. Speed is also a critical parameter of the process; therefore, in the experiment (−1), central (0), and (+1) were 800, 1000, and 1200 rpm, respectively. The measurement of Z-ave was taken after 24 h of preparation (Table 2).

According to Table 2, it is possible to observe that the maximum and minimum values of Z-ave were 748.4 ± 47.6 nm and 578.9 ± 12.3 nm, respectively, and that IP values were less than 0.2. For this optimization, the analysis of variance (Table 3) showed that the quadratic model was significant (*p*-value less than 0.05; α < 0.05) and presented a nonsignificant value for lack of fit (*p*-value = 0.154). The variables showed significant quadratic terms: besifloxacin*besifloxacin, Povacoat^®^*Povacoat^®^, speed*speed. In addition, interactions between besifloxacin and Povacoat^®^ and besifloxacin and rate were observed. The interaction between besifloxacin and Povacoat^®^ reduced particle size, while the interaction between besifloxacin and velocity favored its increase (Equation (1)). Additionally, the quadratic terms influence the Z-ave (increasing). These impacts are also shown in the Z-ave contour plots (Figure 3).

The regression equation obtained for the mathematical model of the experiment is described in Equation (1).
(1)$$\begin{array}{ll} Z-\mathrm{ave}=1608 & -637\;\mathrm{Besifloxacin}-689\;\mathrm{Povacoat}®-1.021\;\mathrm{Speed}+1100\;\mathrm{Besifloxacin}*\mathrm{Besifloxacin}\\ & +790\;\mathrm{Povacoat}®*\mathrm{Povacoat}®+0.000383\;\mathrm{Speed}*\mathrm{Speed}-1200\;\mathrm{Besifloxacin}*\mathrm{Povacoat}®\\ & +0.370\;\mathrm{Besifloxacin}*\mathrm{Speed}. \end{array}$$
where Z-ave: average hydrodynamic diameter.

The coefficient of determination and the adjusted coefficient of determination (R^2^ = 98.37% and R^2^ (Adj.) = 96.16%) are close, allowing the conclusion that nonsignificant values were not considered for the model (Table 4). The predicted R-squared indicates how well a regression model predicts responses to new observations. In this case, the R^2^ (pred) was 75.85%. Generally, values above 70% indicate an excellent forecast.

#### 3.2.2. Mathematical Model Verification

Two formulations with target particle sizes were prepared (F1: target 600 nm, F2: target 576 nm). The same concentration of the reference product was used in this study (0.6% m/m), which is essential to guarantee the minimum inhibitory concentration (MIC). For the formulations, the same conditions were maintained: 30% zirconium spheres and two magnetic bars, with the total amount adjusted to 10 g with phosphate buffer (pH 7.0). The experimental Z-ave (F1 and F2) are close to the theoretical values of each formulation. Therefore, they show the validity of the model (Table 5). Thus, the besifloxacin nanosuspension 0.6 wt% (F2) was selected for posterior analysis due to its reduced particle size (560.3 ± 4.2 nm).

### 3.3. Distribution and Particle Size by Laser Diffraction (LD)

Nanotechnology-based materials or final products are those designed to exhibit properties or phenomena, including physical, chemical, or biological effects that are attributable to its dimensions, even if they are smaller than 1 µm (1000 nm) in size [49].

Given the importance of particle size in nanosuspensions, one of the main characteristics of nanoscale products [50], it is essential to verify particle size by two different techniques. Therefore, a new batch of besifloxacin nanosuspension 0.6 wt% was prepared and analyzed by laser diffraction (LD) and dynamic light scattering (DLS), the latter being the primary technique used during this study. The determination by DLS is related to the Brownian motion of the particles, which promotes the scattering of laser light at different intensities. Consequently, the particle size was determined using the Stokes–Einstein relationship. The particle size distribution by LD is related to the modification of the scattered light intensity as a function of the scattering angle, based on Mie’s theory [31,51].

A saturated drug solution was used for dispersion in the DLS technique and phosphate buffer (pH 7.0) for the LD technique. These precautions are essential to maintain the physical characteristics of the nanocrystals during measurements [42].

The DLS analysis showed Z-ave = 531.1 ± 2 .7 nm and PdI = 0.102 ± 0.031 (Appendix A), while the LD analysis showed D90 = 0.309 ± 0.003 µm, where 95.08% of the particles have a size ≤0.564 ± 0.005 µm, as shown in Figure 4. Thus, the nanosuspension of besifloxacin 0.6% wt% has a particle size approximately 17 times smaller compared to the average particle size of besifloxacin hydrochloride (raw material) and the commercial product (Besivance^®^) (approximately 10 µm). Furthermore, the similarity between the results of the two techniques (DLS and DL) showed that the milling process was adequate.

### 3.4. pH, Density, and Viscosity

The pH for the optimized formulation (besifloxacin nanosuspension 0.6 wt%) was 6.52 using phosphate buffer (pH = 7.0) as diluent. In this condition, the drug is not solubilized, unlike the manufacturing process of the reference product (Besivance^®^). In this case, besifloxacin hydrochloride is dissolved, and after dissolution, the drug is precipitated at pH 6.5 (zwitterion form of the active) [52]. Thus, more steps are required to obtain the commercial product than the besifloxacin nanosuspension.

Therefore, this pH value (pH = 6.5) is within the reference of the United States Pharmacopoeia [53], which establishes that pH between 3.0 and 8.6 is tolerable for ophthalmic formulations. Furthermore, this result can be considered ideal without causing discomfort to patients since tear pH ranges from 6.5 to 7.6, with a mean value of 7.0 [54]. Still, the nanosuspension is suitably buffered, avoiding pH fluctuations and contributing to its stability.

The density of nanosuspension was 1.0081 g/mL, with an average viscosity of 2.40 ± 0.01 cP. These values are within the range of tear fluid viscosity, varying from 1.05 to 5.97 cP. In addition, preparations with a viscosity below 20 cP do not cause discomfort to patients [55]. Determining the viscosity of ophthalmic preparations is not a compendial requirement but must be considered for product specifications, and depends on the proposed formulation [53].

Adequate viscosity allows better Z-ave stability, avoiding aggregation/agglomeration. The Povacoat^®^ is likely adsorbed on the nanocrystal surface, reducing the risk of agglomeration of nanoparticles related to the Ostwald ripening phenomenon [22,56]. In addition, the increase in viscosity favors an increase in the residence time of the drug in the eye, improving bioavailability. However, high viscosity can inhibit the diffusion of drugs from the formulation to the eye [53]. Additionally, high viscosity can increase patient discomfort (pain and blurred vision) and result in dose variability [17]. In the case of the commercial product (Besivance^®^), DuraSite™ technology offers a viscosity of approximately 1500 cps [14].

### 3.5. Osmolality

The mean osmolality obtained for the besifloxacin nanosuspension (0.6 wt%) was 164.3 ± 1.5 mOsmol/kg. The United States Pharmacopeia establishes that the default value for the isosmotic solution is 0.9 wt% sodium chloride (NaCl) solution, which has an osmolality of 285 mOsmol/kg and an osmolarity of 284 mOsmol/L [57]. Therefore, it was proposed to adjust the osmolality of the nanosuspension of besifloxacin 0.6 wt% (optimized formula). Usually, osmolality adjustment can be performed with 0.9 wt% sodium chloride solution; however, this is not recommended for the adjustment of nanosuspensions, as it can change the zeta potential and destabilize the formulation. For this, glycerol was considered, which is a neutral osmotic agent [58].

After adding 1.0 wt% of glycerol, an average osmolality of 303.7 ± 0.6 mOsmol/kg was obtained, close to the typical values of 0.9% sodium chloride solution, as recommended by the United States Pharmacopoeia. Values between 171 and 1711 mOsmol/kg can be tolerated without discomfort to the patient [57]. In addition, these values are like those declared for the reference product Besivance^®^ (290 mOsmol/kg) [5,14].

### 3.6. Validation of the Method by High Performance Liquid Chromatography (HPLC)

For specificity, a placebo containing Povacoat^®^ (stabilizer), glycerol (isotonizing agent), and benzalkonium chloride (preservative), used in the optimized formulation, did not interfere with the analytical responses. In addition, a test was performed using Millex^®^-HV-PVDF filters 13 mm in diameter, with 0.45 µm pores, which had no influence on the analytical responses from the filtering of the solutions.

The method was linear, presenting a correlation coefficient of r = 0.9996 (R^2^ = 99.96%) for drug concentrations ranging from 20 to 80 µg/mL. In the precision by repeatability (intraday), the following values of relative standard deviation (RSD) were obtained: 1.20% on the first day, 0.84% on the second day, and 0.39% on the third day. Regarding intermediate precision, the analyses provided an RSD of 0.94%. The accuracy was evaluated by the recovery method (addition of the standard) and was shown to be satisfactory (99.03%). Therefore, the method is adequate, like the one validated by COSTA and colleagues [12], meeting the recommendations of the ICH guide and United States Pharmacopeia [32,33].

### 3.7. Scanning Electron Microscopy (SEM)

Scanning electron microscopy (SEM) showed the morphology of the raw material besifloxacin (Figure 5a) and the besifloxacin nanocrystals (Figure 5b). Microphotographs showed a change from the irregular prismatic crystal to an approximately spherical shape after the milling process. Furthermore, the micrographs showed that the nanocrystals are involved in a light layer, likely related to the Povacoat^®^ stabilizer [42].

### 3.8. Lyophilization

The lyophilized samples were used for the analyses that needed dry samples. The samples before lyophilization showed Z-ave = 510.5 ± 8.5 nm, PdI = 0.052 ± 0.060 and PZ = −9.66 ± 0.27. After the drying process, these characteristics did not change, presenting Z-ave = 514.4 ± 27.0 nm, PdI = 0.153 ± 0.118, and PZ = −8.66 ± 0.68. Therefore, the lyophilization method was adequate and did not change the characteristics of the nanocrystals. In addition, we verified that the redispersion time of the nanocrystals was fast, and no apparent aggregates were observed. Unlike other lyophilization processes [42], the besifloxacin nanocrystals were successfully dried without adding a cryoprotectant substance. Avoiding the need to add substances is essential due to the restriction of using substances in ophthalmic preparations [59].

### 3.9. Thermal Analysis

The differential scanning calorimetry (DSC) (Figure 6) showed the expected thermal temperature profile for each formulation component analyzed individually. Regarding besifloxacin (raw material), no thermal events were observed in the temperature range studied, consistent with Santos and colleagues [17].

For the nanocrystals, an endothermic curve was observed at approximately 180 °C; in this case, it shifted at a previous temperature with the mixture (197 °C). However, it is possible to show that this curve is related to the sum of the thermal events of the individual components, mainly Povacoat^®^ (Figure 6). Therefore, the analysis showed the thermal stability of the nanocrystals, since no degradation events were observed up to 180 °C. Furthermore, the similarity of the DSC curves of the nanocrystals and the mixture shows that the crystalline form was probably maintained. Additionally, this guarantees the compatibility of the components.

### 3.10. X-ray Diffraction (XRD)

According to the XRD obtained for the nanocrystals (Figure 7), it is possible to observe the prominent peaks of besifloxacin in the 2Θ angles of 10.6, 19.7, and 21.1°, revealing that the crystalline form is likely predominant in the lyophilized formulation. The besifloxacin peaks are also shown in Appendix A. Similar results were presented by Harry [60], where besifloxacin hydrochloride is characterized by diffractograms with peaks at 2Θ angles of 10.6, 15, 19.7, 21.1, and 22° ± 0.2.

Regarding the peaks formed after the angle 2Θ 30°, these are mainly related to the free base of besifloxacin, which present low solubility at pH 6.5, condition of the milling process, and nanosuspension. The same characteristics were observed during the manufacturing process of the Besivance^®^ product, where the HCl salt is converted into the free base [60]. Still, these peaks may be related to the interaction of the drug’s free base with the Povacoat^®^ stabilizer (Figure 7) for the formation of the nanocrystal. Therefore, the crystalline phase is possibly predominant in the nanosuspension.

### 3.11. Saturation Solubility

Analyses to determine the saturation solubility were conducted under 4 pH conditions (ultrapure water, phosphate buffer pH 4.0, pH 6.0, and pH 7.0) [35]. The concentration of besifloxacin dissolved in each medium was determined by HPLC, using the validated method for quantification and indicative of stability. The tests were performed in three replicates with a theoretical concentration of 50 µg/mL for each condition and showed a relative standard deviation below 1.5% for the media, meeting the Brazilian regulatory requirement [35].

Figure 8 shows the saturation solubility of besifloxacin hydrochloride (raw material) and nanosuspension. Regarding the incubation time, no significant changes were observed after 24 h (Figure 8a) and 48 h (Figure 8b). For nanocrystals, the solubility increases approximately twice in phosphate buffer at pH 6.8 and 7.0, relative to the raw material. This performance allows to increase the bioavailability of the drug substance [21,22,23,24,42]. The decrease in the particle radius with a consequent increase in the curvature of their surface causing an increase in the dissolution pressure, according to the Ostwald-Freundlich equation [25]. The reference product (Besivance^®^) presents the same pH range [52]. Furthermore, this is the ideal pH range for ophthalmic formulations, considering the tear pH [54].

The solubility of the raw material in water follows the literature (0.143 mg/mL) [5]. However, the nanosuspension showed lower solubility in this medium (11.11 ± 0.43 µg/mL). This is due to the increase in pH to neutrality when the nanosuspension (buffered at pH 7.0) is added to this medium at the beginning of the tests (Figure 8c). In buffer pH 4.0, raw material and nanosuspension showed solubility 50.65 ± 0.94 µg/mL and 51.50 ± 0.19 µg/mL, respectively.

### 3.12. Determination of Minimum Inhibitory Concentration (MIC)

The MIC values were determined as a function of the presence/absence of microbial growth for the test tubes containing the serial dilutions (0.125 to 60 µg/mL). The MIC for besifloxacin hydrochloride (raw material) and the nanosuspension against *Pseudomonas aeruginosa* was 1.60 µg/mL. Against *Staphylococcus aureus*, the MIC for besifloxacin (raw material) and the nanosuspension were 0.0958 and 0.0960 µg/mL, respectively. These values are like those observed in the preliminary analyses of the raw material and the product (item 3.3). Furthermore, they agree with the literature [8,29,39]. Thus, the milling process did not interfere with the in vitro antibiotic efficacy. Similar results were reported for different nano-based besifloxacin delivery: liposome [17], nanoemulsion [18], nanostructured lipid carrier [19], and nanofibrous [20] (Appendix A).

### 3.13. In Vivo Toxicity

Larvae of *Galleria mellonella* (*n* = 16) were injected using a maximal dose of besifloxacin nanosuspension, Povacoat^®^, and the phosphate buffer (vehicle control). Besifloxacin nanosuspension (500 mg/kg) and the Povacoat^®^ (750 mg/kg) did not show toxicity (Appendix A). The three groups presented 100% larvae survival at the end of the experiment (5 days). An in vivo toxicity study in this model has been used [36,61], mainly because conventional in vitro methods show limitations that make it difficult to simulate physiological responses [62]. Furthermore, this method can support preliminary decisions, reducing the number of tests in mammals [63].

### 3.14. Stability Study

An accelerated stability study was carried out in order to evaluate possible physicochemical changes in the nanosuspension, and samples were incubated at 40 °C ± 2 °C/75% RH ± 5% RH. As for the aspect, evaluation of the samples carried out at each time point of the study indicate homogeneity, without change in color, odor, phase separation, and significant sample sedimentation. At each moment, slight sediment of approximately 1 mm in thickness was seen at the bottom of the flasks; this sediment is broken up easily through manual homogenization.

About Z-ave, an increase above 50 nm was observed between the initial analysis (time 0) and 30 days for the three batches (Figure 9). Therefore, the analysis of variance (ANOVA) showed a statistically significant change (*p*-value = 0.00033; α = 0.05) for the period evaluated. This may be related to the small sediment (≈1 mm thick) observed in the visual analysis. However, in the later periods, Z-ave stability was observed, and between 30 and 90 days, there was no statistically significant difference (*p*-value = 0.94; α = 0.05). Therefore, its stability can be improved by increasing the viscosity of the formulation and/or adding components that increase the steric hindrance of the formulation, such as ammonium quaternary.

Additionally, a monomodal distribution was observed (Appendix A) after 90 days of incubation, with PdI ≤ 0.2, and no statistically significant change (*p*-value = 0.22239; α = 0.05) (Figure 9). Similarly, no significant changes were observed in pH (*p*-value = 0.78369; α = 0.05) and zeta potential (*p*-value = 0.18212; α = 0.05) (Figure 10). Additionally, the content was stable at the end of the period.

## 4. Conclusions

The besifloxacin nanocrystals were successfully prepared using small-scale bed milling and response surface methodology. Povacoat^®^ 0.9 wt% reduced the particle size of besifloxacin hydrochloride (550 nm) 17 times, which increased the saturation solubility by approximately two times concerning the raw material. The besifloxacin nanocrystal 0.6 wt% showed in vitro efficacy and did not show toxicity in a larval model. In addition, it presented satisfactory physical and chemical stability, and was able to meet the requirements of a commercial product. Thus, besifloxacin nanocrystals revealed the potential for reduced doses of the drug substance, with the minor occurrence of adverse effects and greater patient adherence to treatment.

## Figures and Tables

**Figure 1 pharmaceutics-14-02221-f001:**
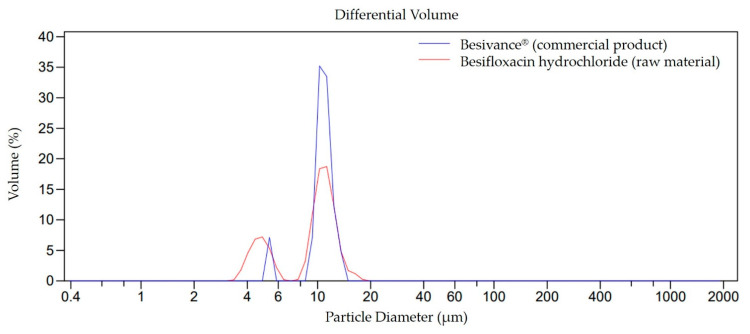
Particle size distribution for the commercial product Besivance^®^ and besifloxacin hydrochloride. The peaks represent the average of three readings.

**Figure 2 pharmaceutics-14-02221-f002:**
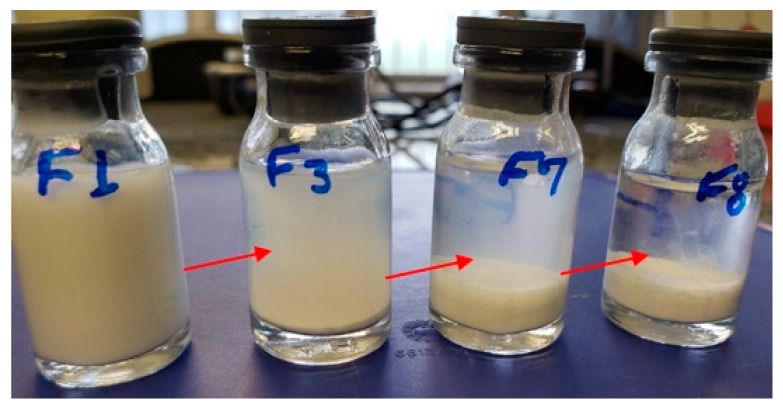
Aspect of the formulas containing the stabilizers Povacoat^®^ 1.0 wt% (F1), Kollicoat^®^ 0.3 wt% + Chitosan 0.3 wt% (F3), Poloxamer^®^ 407 0.5 wt% + Chitosan 0.5 wt% (F7), and Poloxamer^®^ 407 0.3 wt% + Chitosan 0.3 wt% (F8). Stability after 15 days (25 °C).

**Figure 3 pharmaceutics-14-02221-f003:**
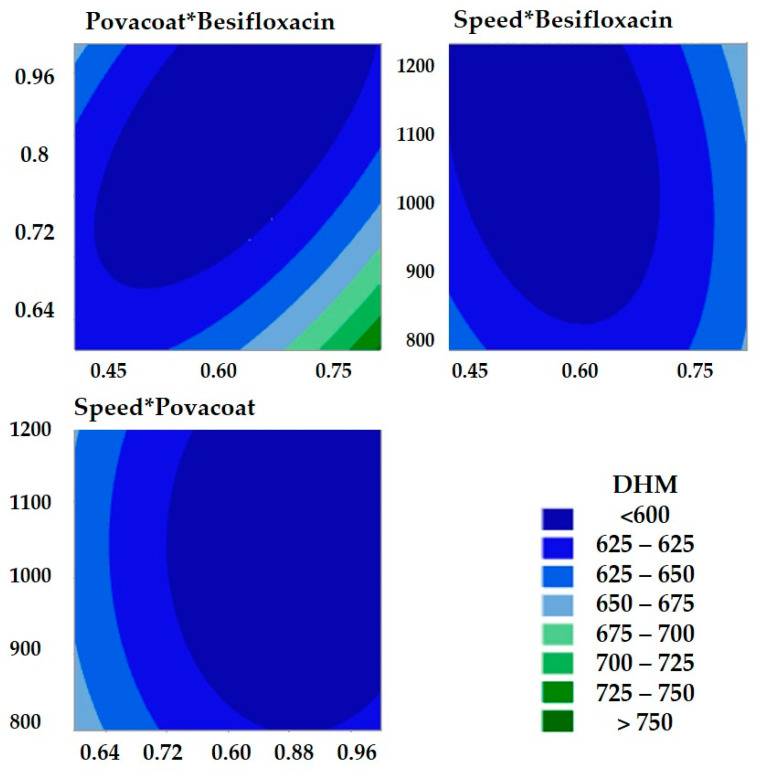
Contour graph related to the Z-ave evaluation of the besifloxacin nanocrystal, whose variables are besifloxacin concentration (wt%), Povacoat^®^ concentration (wt%), and velocity (rpm).

**Figure 4 pharmaceutics-14-02221-f004:**
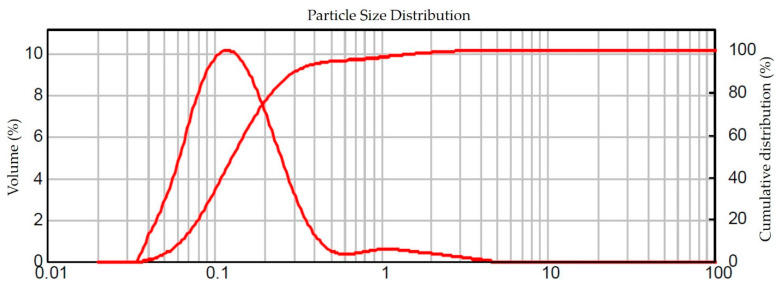
Particle size distribution of besifloxacin nanosuspension 0.6 wt% by laser diffraction (LD). Representative of three runs (*n* = 3).

**Figure 5 pharmaceutics-14-02221-f005:**
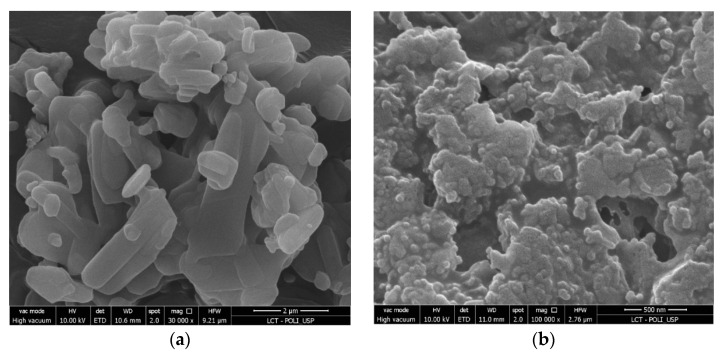
Microphotography of raw material besifloxacin (**a**) and besifloxacin nanocrystals (**b**).

**Figure 6 pharmaceutics-14-02221-f006:**
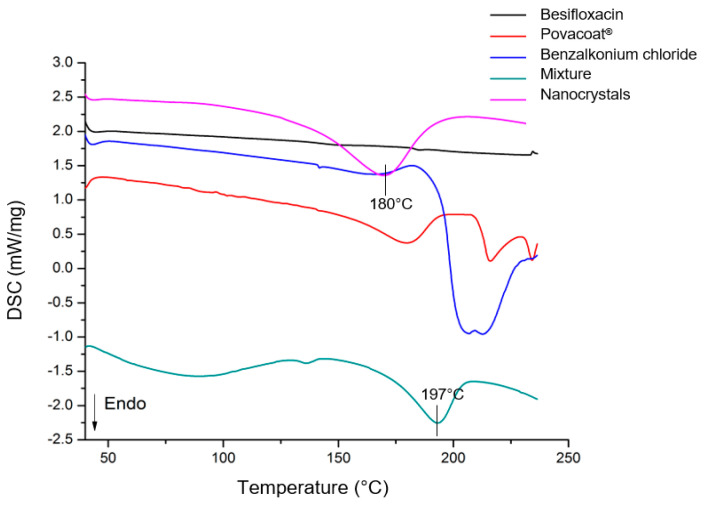
DSC curves of formulation components, mixture (same proportion as formulation), and besifloxacin nanocrystals.

**Figure 7 pharmaceutics-14-02221-f007:**
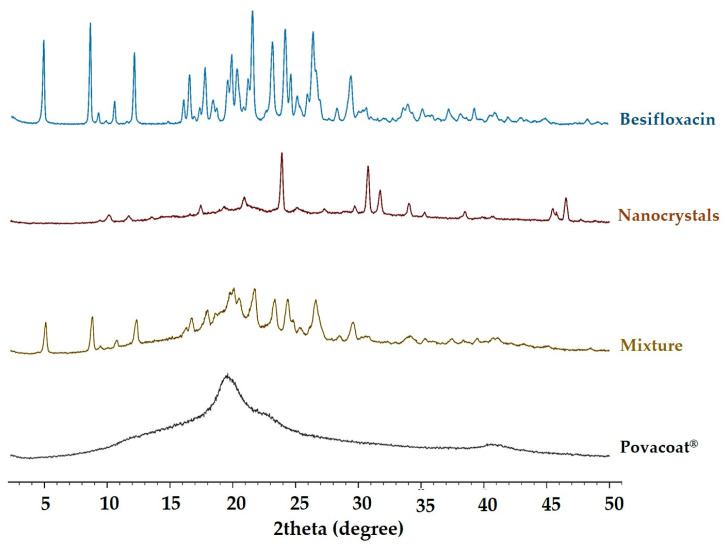
X-ray diffraction of besifloxacin (raw material), nanocrystals, mixture (besifloxacin and Povacoat^®^), and Povacoat^®^. *n* = 3.

**Figure 8 pharmaceutics-14-02221-f008:**
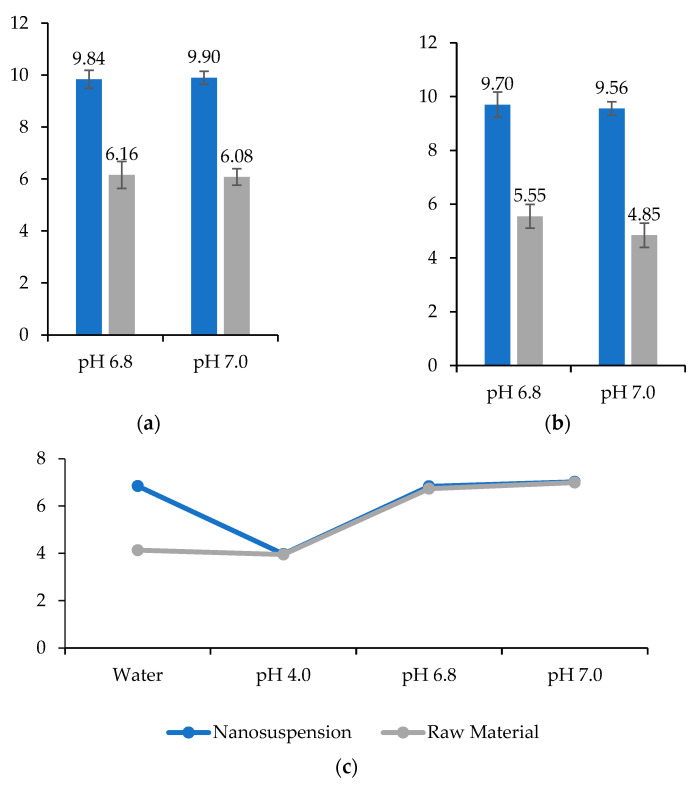
Saturation solubility of besifloxacin (raw material) and nanosuspension (besifloxacin 0.6 wt%). (**a**) Solubility after 24 h of incubation. (**b**) Solubility after 48 h of incubation. (**c**) pH of the samples at the end of the incubation. Theoretical concentration: 50 µg/mL. *n* = 3.

**Figure 9 pharmaceutics-14-02221-f009:**
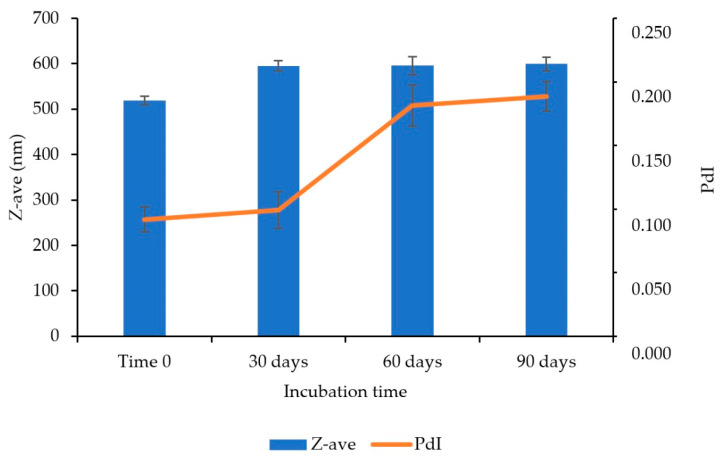
Determination of the average hydrodynamic diameter (Z-ave) in nm and polydispersion index (PdI), for the accelerated stability study of the nanosuspension of besifloxacin 0.6 wt%, in the time intervals: T0, 30, 60, and 90 days. Study condition: 40 °C ± 2 °C/75% RH ± 5% RH. *n* = 3 batch.

**Figure 10 pharmaceutics-14-02221-f010:**
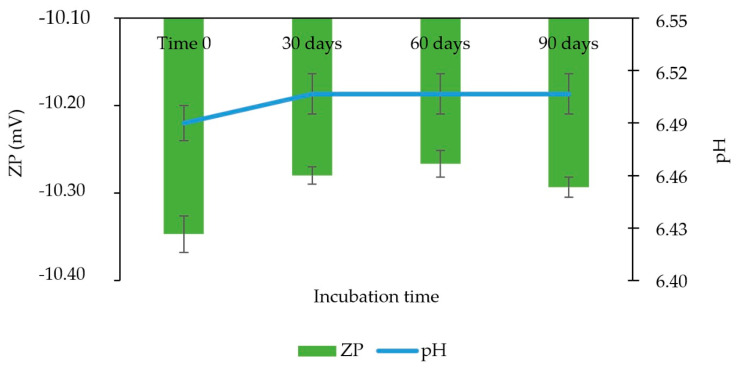
Determination of zeta potential (ZP) and pH for the accelerated stability study of the nanosuspension of besifloxacin 0.6 wt%, in the time intervals: T0, 30, 60, and 90 days. Study condition: 40 °C ± 2 °C/75% RH ± 5% RH. *n* = 3 batch.

**Table 1 pharmaceutics-14-02221-t001:** Independent variables and levels selected for Box–Behnken design.

Level	Besifloxacin (wt%)	Stabilizer (Povacoat^®^) (wt%)	Stirring Speed (rpm)
Minimum (−1)	0.4	0.6	800
Central point (0)	0.6	0.8	1000
Maximum (+1)	0.8	1.0	1200

**Table 2 pharmaceutics-14-02221-t002:** Box–Behnken experimental design and observed responses. Average hydrodynamic diameter (Z-ave) in nm was obtained for the besifloxacin nanocrystal preparations. Conditions: 30% (m/m) zirconium beads and 2 magnetic bars, 25 °C.

Formula	Order	Central Point	Besifloxacin (wt%)	Povacoat^®^ (wt%)	Speed (rpm)	Z-ave (nm)	PdI
1	1	2	0.4	0.6	1000	641.2 ± 45.2	0.110 ± 0.123
2	12	2	0.6	1.0	1200	596.7 ± 27.6	0.199 ± 0.137
3	11	2	0.6	0.6	1200	651.7 ± 61.8	0.131 ± 0.174
4	8	2	0.8	0.8	1200	677.5 ± 78.4	0.145 ± 0.137
5	5	2	0.4	0.8	800	638.6 ± 33.0	0.204 ± 0.166
6	13	0	0.6	0.8	1000	585.3 ± 21.7	0.168 ± 0.069
7	4	2	0.8	1.0	1000	592.9 ± 28.1	0.199 ± 0.187
8	10	2	0.6	1.0	800	608.3 ± 18.6	0.198 ± 0.186
9	2	2	0.8	0.6	1000	748.4 ± 47.6	0.183 ± 0.107
10	7	2	0.4	0.8	1200	595.4 ± 35.2	0.139 ± 0.070
11	3	2	0.4	1.0	1000	666.7 ± 37.6	0.310 ± 0.217
12	9	2	0.6	0.6	800	666.7 ± 51.6	0.197 ± 0.141
13	6	2	0.8	0.8	800	661.5 ± 35.4	0.162 ± 0.104
14	15	0	0.6	0.8	1000	587.6 ± 30.1	0.110 ± 0.103
15	14	0	0.6	0.8	1000	578.9 ± 12.3	0.158 ± 0.039

**Table 3 pharmaceutics-14-02221-t003:** Analysis of variance to test the significance of the terms obtained for the particle Z-ave variation of the besifloxacin nanocrystal preparations. The variables are besifloxacin concentration (wt%), Povacoat^®^ concentration (wt%), and velocity (rpm).

Source	DF	Adj. SS	Adj. MS	F-Value	*p*-Value
Model	8	30,471.9	3808.99	45.16	0.0001
Linear	3	9902.4	3300.82	39.14	0.0001
Besifloxacin	1	2791.2	2791.17	33.09	0.001
Povacoat^®^	1	6749.5	6749.48	80.03	0.0001
Speed	1	361.8	361.81	4.29	0.084
Square	3	10,474.4	3491.47	41.40	0.0001
Besifloxacin * Besifloxacin	1	7149.8	7149.80	84.77	0.0001
Povacoat^®^ * Povacoat^®^	1	3688.1	3688.06	43.73	0.001
Speed * Speed	1	865.7	865.70	10.26	0.019
Interaction between 2 factors	2	10,095.0	5047.52	59.85	0.0001
Besifloxacin * Povacoat^®^	1	9218.9	9218.88	109.30	0.0001
Besifloxacin * Speed	1	876.2	876.16	10.39	0.018
Error	6	506.1	84.34		
Lack-of-fit	4	465.4	116.35	5.72	0.154
Pure error	2	40.6	20.32	*	*
Total	14	30,977.9			

DF: degree of freedom; Adj. SS: Sum of the adjusted squares; Adj. MS: Adjusted quadratic mean, F-value: F statistic; *p*-value: significance level; (*): not calculated. Besifloxacin square (Besifloxacin * Besifloxacin); Povacoat® square (Povacoat® * Povacoat®); Speed square (Speed * Speed); interaction between Besifloxacin and Povacoat® (Besifloxacin * Povacoat®), and interaction between Besifloxacin and Speed (Besifloxacin * Speed).

**Table 4 pharmaceutics-14-02221-t004:** Significance of coded regression coefficients and fit index of the selected model in evaluating Z-ave nanocrystals of besifloxacin.

Term	Coef	SE Coef	t-Value	*p*-Value	VIF
Constant	583.93	5.30	110.13	0.0001	
Besifloxacin	18.68	3.25	5.75	0.001	1.00
Povacoat^®^	−29.05	3.25	−8.95	0.000	1.00
Speed	−6.72	3.25	−2.07	0.084	1.00
Besifloxacin * Besifloxacin	44.00	4.78	9.21	0.0001	1.01
Povacoat^®^ * Povacoat^®^	31.60	4.78	6.61	0.001	1.01
Speed * Speed	15.31	4.78	3.20	0.019	1.01
Besifloxacin * Povacoat^®^	−48.01	4.59	−10.45	0.0001	1.00
Besifloxacin * Speed	14.80	4.59	3.22	0.018	1.00
S: 9.18% R^2^: 98.37% R^2^ (adj.): 96.16% R^2^ (pred): 75.85%					

Coef: Coefficient; SE Coef: Standard deviation coefficient; *t*-value: size of the difference relative to the variation; *p*-value: level of significance of terms; VIF: variance inflation factor; S: standard deviation; R^2^: coefficient of determination; R^2^ (adj.): Adjusted coefficient of determination; R^2^ (pred): Coefficient of prediction determination of the adjusted model. Besifloxacin square (Besifloxacin * Besifloxacin); Povacoat® square (Povacoat® * Povacoat®); Speed square (Speed * Speed); interaction between Besifloxacin and Povacoat® (Besifloxacin * Povacoat®), and interaction between Besifloxacin and Speed (Besifloxacin * Speed).

**Table 5 pharmaceutics-14-02221-t005:** Variables obtained in the Box–Behnken experimental design for the optimized nanocrystal preparations and respective theoretical and experimental Z-ave values.

Formula	Besifloxacin (wt%)	Povacoat^®^ (wt%)	Speed (rpm)	Theoretical Z-ave (nm)	Experimental Z-ave (nm)	PdI	ZP (mV)
F1	0.6	0.8	834	600	639.9 ± 30.3	0.101 ± 0.056	−10.10 ± 0.15
F2	0.6	0.9	1044	576	560.3 ± 4.2	0.098 ± 0.067	−10.20 ± 0.10

## Data Availability

The data presented in this study are available on request from the corresponding author.

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
