# Peer review of "Besifloxacin Nanocrystal: Towards an Innovative Ophthalmic Preparation"

_pharmaceutics, 2022, doi:10.3390/pharmaceutics14102221_

Round 1
Reviewer 1 Report
Besifloxacin Nanocrystal: Towards An Innovative Ophthalmic Preparation
Comments to the authors:
PXRD figure of the Figure 7 need to redraw. It not clear to discuss PXRD lines and X, Y axis scales also need to change.
Authors need to make a table in SI to show what are previous reports of current drug, drug salt etc, and what is different here and what is novelty in detail, which is missing.
Indeed, Authors need to make a table where it can show nanocrystals of the drugs are better than normal powder of macro size, though it known in literature, here it is missing to understand.
Reviewer 2 Report
This work focus an important topic, of interest for the readers of Pharmaceutics. However, a major revision is needed, addressing the comments below, and an additional experiment must mandatorily be performed and added to the manuscript.
1. The Introduction must be more focused (some parts like the origin/fabrication of the chosen antibiotics are completely useless), stating clearly the innovation of the work. This is not very clear.
2. Lines 115.118: No sense
3. DLS or laser diffraction: Correlograms must be included in the manuscript, together with the corresponding graphs of size distribution. Moreover, at least five measurements should be performed. Figure 4 has only one measurement (and no correlation curve).
4. Figures’ quality must be improved. Several figures do not have enough quality for publication, namely Figure 1, Figure 3, Figure 4, Figure 6, Figure 7.
5. Lines 339-340: “Therefore, the raw material besifloxacin hydrochloride is presented on a micrometric scale, close to the reference values (commercial product). Thus, they justify the process of obtaining drug nanocrystals by small-scale wet bead milling.” Why? Please explain this statement.
6. Sections 3.1.2. and 3.12: MIC values presented do not have Standard Deviation. How many measurements were performed to obtain statistical significance? Only one measurement is not acceptable.
7. Equation 1 is unformatted. Please, format equation properly.
8. Some statements in the manuscript are out of sense, like the definition of “a nanotechnology-based product” (lines 459-463).
9. XRD figure 7: Axis titles need correction.
10. Line 594: What is the meaning of “BRASIL, 2011”?
11. Reference 55 needs reformatting
12. Stability studies must be performed at 35 ºC, the normal temperature of corneal surface and not at 40 ºC. In fact, 40 ºC is equivalent to a real situation of high fever. High temperatures may enhance nanocrystals and nanoemulsions’ stability (unless the drug was temperature-sensitive), giving rise to better results than those in a real situation. Please, add these studies at 35 ºC in comparison with the ones at 40 ºC. This is a mandatory additional experiment.
Reviewer 3 Report
The novel formulation is a promising candidate for the development of a generic product.
I recommend the introduction of a graphical abstract in order to increase accessibility for readers.
Also, concerning the determination of MIC it would have been interesting to compare the values obtained for the commercial product already available Besivance®, 0.6% ophthalmic suspension.
Reviewer 4 Report
Overall, Besifloxacin Nanocrystal: Towards An Innovative Ophthalmic Preparation is an excellent manuscript. The authors have done a great job in writing this interesting, novel research work in a strong, clear, and organized manner with proper flow.
Please address the following comments.
Comment 1: Line 119-120 “A suspension of the drug was prepared in pH 7.0 buffer, and the commercial product (Besivance®) was used directly for analysis” which buffer phosphate, acetate?
Comment 2: line 214 “Measurements were taken in triplicate” it is better to use performed or conducted instead of taken.
Comment 3: Could you please provide polydispersity index values with the particle size data if possible.
Round 2
Reviewer 2 Report
The comments of the reviewers were addressed.